# Healthcare workers' experiences with COVID-19-related prevention and control measures in Tanzania

**Kassimu Tani**[1,2,3]*, **Brianna Osetinsky**[2,3], **Grace Mhalu**[1], **Sally Mtenga**[1], **Günther Fink**[2,3], **Fabrizio Tediosi**[2,3]

**1** Department of Health System, Impact Evaluation and Policy, Ifakara Health Institute, Dar es Salaam, Tanzania, **2** Swiss Tropical and Public Health Institute, Basel, Switzerland, **3** University of Basel, Basel, Switzerland

* ktani@ihi.or.tz

**Data Availability Statement:** The datasets used and/or analysed during the current study are available from the corresponding author on reasonable request. A data transfer agreement will

## Abstract

The ability of a health system to withstand shocks such as a pandemic depends largely on the availability and preparedness of health-care workers (HCWs), who are at the frontline of disease management and prevention. Despite the heavy burden placed on HCWs during the COVID-19 pandemic, little is known regarding their experiences in low-income countries. We conducted a web-based survey with HCWs in randomly selected districts of Tanzania to explore their experiences with COVID-19-related prevention and control measures. The survey assessed implementation of COVID-19 control guidelines in health facilities, HCW perceptions of safety, well-being and ability to provide COVID-19 care, and challenges faced by frontline workers during the pandemic. We used multivariate regression analysis to examine the association between HCW and health facility characteristics, a score of guideline implementation, and challenges faced by HCWs. 6,884 Tanzanian HCWs participated in the survey between December 2021 to March 2022. The majority of respondents were aware of the COVID-19 guidelines and reported implementing preventive measures, including masking of both HCWs and patients. However, HCWs faced several challenges during the pandemic, including increased stress, concerns about infection, and inadequate personal protective equipment. In particular, female HCWs were more likely to report exhaustion from wearing protective equipment and emotional distress, while physicians were more likely to experience all challenges. While most HCWs reported feeling supported by facility management, they also reported that their concerns about COVID-19 treatment were not fully addressed. Notably, perceptions of protection and well-being varied widely among different HCW cadres, highlighting the need for targeted interventions based on level of exposure. In addition, various factors such as HCW cadre, facility ownership and COVID-19 designation status influenced HCWs' opinions about the health system's response to COVID-19. These findings highlight the importance of consistent implementation of guidelines and social and emotional support for HCWs.

be signed between the Ifakara health institute (IHI) and Swiss TPH and the requesting institution before data sharing. The ethical approvals from the Ethical Committee at the National Institute for Medical Research (NIMR) in Tanzania, the Ifakara Health Institute's (IRB) do not give us a license to make the dataset publicly available. In case of data request, please contact the IRB secretary, Dr. Mwifadhi Mrisho (+255655766675, mmrisho@ihi. or.tz).

**Funding:** This study was supported by Swiss Programme for Research on Global Issues for Development (r4d programme) grant number 183760, a joint funding initiative by the Swiss Agency for Development and Cooperation (SDC) and the Swiss National Science Foundation (SNSF). FT received this award. https://p3.snf.ch/project-183760 https://www.snf.ch/en https://www.eda. admin.ch/sdc. The funders had no role in study design, data collection and analysis, decision to publish, or preparation of the manuscript.

**Competing interests:** The authors have declared that no competing interests exist.

## Introduction

Shortages of health care workers (HCWs) are common in low-income countries, and the available HCW often face challenging working conditions and limited resources [1–3]. The ability of a health system to withstand adverse events, such as a pandemic, depends largely on the availability and preparedness of HCWs [4–6]. However, many countries struggle to have enough HCWs in all health facilities to meet existing and emerging health needs [3,7–11]. During the COVID-19 pandemic, HCWs were on the frontline, facing both personal risks and changes in their work environment to help contain the spread of the disease and manage patients [12–15]. Several studies have shown that the pandemic has placed a heavy burden on HCWs [13,16], with factors such as institutional decisions, organisational support and government responses influencing their experiences [17–19]. A country's ability to manage health and other competing priorities during a pandemic depends largely on the responsiveness of the health system and its ability to rapidly develop and implement strategies that can be implemented and supported by HCWs [5,20–23]. The effectiveness of interventions therefore depends not only on the timely adoption of evidence-based interventions, but also on how these interventions are implemented and perceived by frontline workers [24].

The Tanzanian government under the president John Magufuli initially took a controversial approach to COVID-19, downplaying the severity of the pandemic and resisting many of the measures recommended by health experts. However, in March 2021, the new president, Samia Suluhu Hassan, announced a change in approach and formed a committee to investigate the country's COVID-19 situation. In response to the COVID-19 pandemic, health facility managers have been instructed to improve hygiene infrastructure and increase the availability of personal protective equipment [22,25]. The government implemented public messaging and national campaigns to promote hand washing and to increase the availability of water and hand washing facilities in public places [25–27]. The government issued guidelines to strengthen the health system's response, designated health facilities in each district to isolate and treat COVID-19 patients, and implemented several guidelines to strengthen the production of protective equipment recommended by the World Health Organization (WHO) [19,28,29]. These initiatives were aimed at strengthening the capacity of the health system to deal with emergencies and pandemics, to treat patients and to improve the communication of COVID-19 to reduce hesitation and misinformation [30–32].

Despite the growing body of literature on responses to COVID-19 [33–35], the response of health systems in low-income countries to the pandemic remains poorly understood. Evidence appears to be particularly lacking for experiences and perspectives of HCWs in low-income settings. This study aimed to address this gap by exploring HCWs' experiences of COVID-19-related prevention and control measures implemented in Tanzania.

## Methods

### Study design and setting

This is a cross-sectional survey with data collected from HCWs in selected districts of Tanzania using a web-based questionnaire. The study included all cadres of HCWs such as nurses, doctors, clinical officers who are mid-level providers, laboratory technicians and pharmacists.

### Sampling and data collection

We randomly selected 20% of the districts in Tanzania using a population-weighted probability of sampling. To account for municipalities with separate local councils but the same name as the surrounding district, we combined their populations for sample weighting. This strategy

allowed us to ensure that urban centres, which are often geographically smaller but are more populous, and surrounding rural areas, which share some infrastructural resources including major health facilities, were adequately represented. Our sample consisted of 39 districts out of 147 in 19 regions. From these districts, we then estimated the maximum number of participants to be 17,544, based on the number of health facilities in the selected districts and the number of HCWs expected to work in each facility according to Ministry of Health (MoH) regulations. We used the Open Data Kit (ODK) and Enketo to collect data from all HCWs practicing in the study districts from December 2021 to March 2022. At the district level, the district nursing officer (DNO) and/or district health secretary coordinated data collection by distributing the survey link through local WhatsApp groups of HCWs, and district coordinators provided follow-up and encouragement to increase completion. We reimbursed HCWs for the airtime used to complete the questionnaire with an amount equivalent to USD 2 sent directly to their mobile money, but they were not otherwise compensated.

## Questionnaire and variables description

We developed the questionnaire using an iterative approach. We first reviewed the relevant literature and international and national guidelines on COVID-19 management. We then sought input from senior medical researchers with experience of health services in Tanzania. After this valuable consultation, we submitted the questionnaire to the Tanzanian Ethics Committee, which provided insightful feedback. We incorporated their recommendations to refine the questionnaire. The revised version was then distributed to a selected group of healthcare providers via their WhatsApp numbers to pilot the questionnaire. We considered and responded to the feedback we received, making the necessary adjustments to ensure the effectiveness of the questionnaire.

The HCW questionnaire assessed the implementation of COVID-19 control guidelines in health facilities, HCW perceptions of safety, well-being and ability to provide COVID-19 care, and the challenges they experienced as frontline workers during the COVID-19 pandemic up to the day of the survey. The survey included the following sections.

1. Demographic information about the respondent, such as age, gender, years since graduation, and cadre of HCW.

2. Information on the health facility where the respondent worked most of the time, including health facility location (rural/urban), level, ownership, and information on the availability of COVID-19 services.

3. Questions about awareness of policies or guidelines for COVID-19 and which level of government the guidelines came from. We developed a list of 20 key guidelines that may have been implemented in health facilities, based on available guidelines at national and regional levels for the clinical management and infection prevention and control of COVID-19 [29].

4. Questions about challenges faced by HCWs since the start of the COVID-19 pandemic. We asked if they had experienced violence or threats against HCWs; if they had experienced stress—dealing with the risk of COVID-19, exhaustion from wearing protective gear, fatigue from increased workload, fear of being infected or infecting others, and emotional distress due to powerlessness when patients deteriorate despite their efforts, or due to high mortality.

5. Questions on HCWs' perceptions of the implementation of COVID-19 control and treatment policies (Table A in S1 Appendix). To generate an overall composite score and separate composite scores for three thematic groups of guidelines (general precautions, changes in care protocols to control and prevent COVID-19, and COVID-19 triage and treatment protocols), we summed the responses on implemented guidelines for each question.

We employed a five-point Likert scale encompassing 13 perception-based questions regarding the implementation of COVID-19 control and treatment guidelines. This scale ranged

from 1, indicating 'strongly disagree,' to 5, signifying 'strongly agree.' To maintain uniformity and consistency, we applied reverse coding to questions with negative wording, ensuring that a rating of 5 consistently represented a positive attitude.

In our data analysis, we utilized an overarching perception scale that combined all Likert-style scores. Furthermore, we generated two distinct component scores. The first encapsulated the responses to 7 questions pertaining to the health and well-being of HCWs, while the second encompassed the 6 questions related to COVID-19 treatment and guidelines.

## Data analysis

We tabulated information on the demographics of HCWs and the main health facility where they work, including information on COVID-19 services, the origin of COVID-19 guidelines, and changes in patient volume at the facility since the start of COVID-19. We presented frequency distributions for categorical variables and used chi-squared tests to compare mean responses across health facilities.

Multivariate regressions were conducted to explore the relationship between the implementation of guidelines, challenges encountered by HCWs, and their perceptions of the implementation of COVID-19 control and treatment policies, taking into account specific characteristics of both HCWs and health facilities. The dependent variables under consideration included scores for guidelines implementation, challenges faced by HCWs, and HCWs' perceptions on a Likert scale.

Independent variables included HCW age (grouped into three categories to reflect differential risk of severe COVID-19 infection or death for young adults, middle age adults, and older adults: 18–29, 30–49 and 50+) [36], gender, cadre, health facility level, ownership, rural/urban location, whether facility was designated as an official COVID-19 treatment center, availability of COVID-19 testing, whether facility had treated suspected or confirmed COVID-19 cases, institutions issuing COVID-19 guidelines (MOH, regional, district or local authorities), and whether HCWs were aware of COVID-19 guidelines. We specified three models for each outcome indicator. The first model used was a linear regression for implementation scores. It aimed to measure the relationship between existing predictors of implementation of guidelines for prevention and treatment of COVID-19 within health facilities. The second model was a logistic regression for the Likert scale scores of HCW perception of protection and well-being. The third model was a logistic regression intended to measure the predictors of reported challenges faced by HCWs. Standard errors were clustered at the district level, and all analyses were conducted using Stata IC 16. P value of <0.05 was considered statistically significant.

## Ethics statement

Ethical approval for this study was obtained from the Ifakara Health Institute (IHI) Institutional Review Board (IHI/IRB/EXT/No: 35–2020) and the Tanzanian National Institute for Medical Research (NIMR/HQ/R.8a/Vol.IX/3726). Written informed consent was obtained prior to the start of the survey, and respondents were required to confirm their understanding and intent to proceed by ticking three boxes, providing written confirmation, and signing the smartphone used to complete the survey in order to proceed to the survey. The survey was end-to-end encrypted to ensure respondents' privacy, and data was sent to a secure server at the Ifakara Health Institute.

## Inclusivity in global research

Additional information regarding the ethical, cultural, and scientific considerations specific to inclusivity in global research is included in the Supporting Information (S1 Checklist).

### Role of the funding source

The funders had no role in study design, data collection and analysis, decision to publish, or preparation of the manuscript.

## Results

### Respondent characteristics

The study included 6,884 respondents, or about 39.2% of the estimated total number of HCWs in the targeted districts. The majority (31%) working in dispensaries, followed by health centers (27%), hospitals (23%), referral hospitals (14%) and out-of-referral clinics (5%). Among health workers, 47% were female, with an even distribution across facilities. The majority (56.7%) of HCWs were under 30 years of age and had less than 5 years of experience (42%). Hospitals and referral hospitals had a higher proportion of HCWs over 50 years of age and with more experience. Clinical officers were more common in dispensaries (31%) and health centers (24%), while hospitals (19%) and referral hospitals (38%) had more doctors. Nurses were the largest group of HCWs at all facility levels, accounting for 41% of the total sample, but were more common in dispensaries (47%). Clinics had the highest proportion of technicians (17%), pharmacists (18%) and other HCWs (14%) as they provide specialised services. The majority of HCWs (51%) worked in urban facilities, with higher proportions in hospitals (64%) and referral hospitals (90%) in larger urban centers. Government facilities employed 74% of HCWs, 11% worked in NGO or faith-based facilities, and only 15% worked in private facilities, with the majority of clinic HCWs (86%) working in private facilities (Table 1).

### Covid-19 services and awareness of guidelines availability

Only 45% of clinics provided any COVID-19 treatment, compared to 93% at referral hospitals. 60% of HCWs reported a decrease in patient volume since the start of the COVID-19 pandemic, with a higher proportion of clinics (68%) and referral hospitals (75%) reporting such decreases (Table 1). In terms of guidelines, only a small percentage of HCWs (6%) were not aware of any guidelines related to the response of the health facility to the control and management of COVID-19. However, for HCWs working in dispensaries (9%) and clinics (14%), this percentage was higher. Most HCWs (73%) were aware of MOH guidelines, with 93% of HCWs in referral hospitals knowing that MOH was the source of COVID-19 treatment and control guidelines for health facilities. Knowledge of guidelines from the regional or district level was also reported by HCWs in dispensaries, health centres and hospitals. Of those HCWs working in clinics who were aware of guidelines, 17% reported that they were sourced from the community or village level. It is worth noting that guidelines from different sources were not mutually exclusive, so it was possible for HCWs to be aware of guidelines issued by different levels of government (Table 1).

### Implementation of guidelines to mitigate the health and health systems consequences of COVID-19

The most commonly implemented changes were masking of HCWs and clients, prioritising care for comorbidities, and minimising the number of people accompanying clients (Table 2). There was considerable variation in guideline implementation, ranging from 83% of the total sample reporting masking of HCWs and clients to only 13.5% reporting that routine medical or laboratory visits were postponed in their facility. Implementation also varied between different levels of health facilities, with COVID-19 treatment protocols more commonly reported in

**Table 1. Responders and health facilities characteristics.**

| | Total n (%) | Dispensary n (%) | Health Centre n (%) | Hospital n (%) | Referral Hospital n (%) | Clinics[a] n (%) | P-Value |
|---|---|---|---|---|---|---|---|
| | N = 6,884 | 2,160 (31.4) | 1,851 (26.9) | 1,563 (22.7) | 980 (14.2) | 330 (4.8) | |
| **Gender** | | | | | | | |
| Male | 3,626 (52.7) | 1,157 (53.6) | 957 (51.7) | 823 (52.7) | 513 (52.4) | 176 (53.3) | 0.829 |
| Female | 3,258 (47.3) | 1,003 (46.4) | 894 (48.3) | 740 (47.3) | 467 (47.6) | 154 (46.7) | |
| **Age category** | | | | | | | |
| 18–29 | 3,898 (56.7) | 1,277 (59.2) | 1,036 (56.0) | 784 (50.2) | 535 (54.7) | 266 (80.6) | <0.001 |
| 30–49 | 2,741 (39.9) | 821 (38.0) | 763 (41.3) | 696 (44.6) | 399 (40.8) | 62 (19.8) | |
| 50+ | 239 (3.5) | 61 (2.8) | 50 (2.7) | 45 (4.6) | 81 (5.19) | 2 (0.6) | |
| **Cadre** | | | | | | | |
| Nurse | 2,889 (41.9) | 1,024 (47.4) | 746 (40.3) | 636 (40.7) | 377 (38.5) | 106 (32.1) | <0.001 |
| Clinical officer | 1,366 (19.8) | 668 (30.9) | 440 (23.8) | 196 (12.5) | 31 (3.2) | 31 (9.4) | |
| Medical Doctors | 938 (13.6) | 33 (1.5) | 200 (10.8) | 303 (19.4) | 374 (38.2) | 28 (8.5) | |
| Technicians | 653 (9.5) | 139 (6.4) | 205 (11.1) | 180 (11.5) | 72 (7.4) | 57 (17.3) | |
| Pharmacist | 377 (5.5) | 74 (3.4) | 114 (6.2) | 90 (5.8) | 40 (4.1) | 59 (17.9) | |
| Others | 661 (9.6) | 222 (10.3) | 146 (7.9) | 158 (10.1) | 86 (8.8) | 49 (14.9) | |
| **Years of Experience** | | | | | | | |
| Missing | 1,233 (17.9) | 374 (17.3) | 350 (18.9) | 264 (16.9) | 190 (19.4) | 55 (16.7) | <0.001 |
| < 5 Years | 2,896 (42.1) | 830 (38.4) | 750 (40.5) | 633 (40.5) | 465 (47.5) | 218 (66.1) | |
| 5 to 9 Years | 1,864 (27.1) | 695 (32.2) | 537 (29.0) | 410 (26.2) | 175 (17.9) | 47 (14.2) | |
| 10+ Years | 891 (12.9) | 261 (12.1) | 214 (11.6) | 256 (16.4) | 150 (15.3) | 10 (3.0) | |
| **Health facility location** | | | | | | | |
| Rural | 3,359 (48.8) | 1,590 (73.6) | 977 (52.8) | 571 (36.5) | 95 (9.7) | 126 (38.2) | <0.001 |
| Urban | 3,525 (51.2) | 570 (26.4) | 874 (47.2) | 992 (63.5) | 885 (90.3) | 204 (61.8) | |
| **Health facility ownership** | | | | | | | |
| Government | 5,098 (74.1) | 1,706 (78.9) | 1,508 (81.5) | 1,058 (67.7) | 819 (83.6) | 7 (2.1) | <0.001 |
| NGO/Religious | 766 (11.1) | 158 (7.3) | 150 (8.1) | 305 (19.5) | 112 (11.4) | 41 (12.4) | |
| Private | 1,020 (14.8) | 296 (13.7) | 193 (10.4) | 200 (12.8) | 49 (5.0) | 282 (85.5) | |
| **Designated to offer COVID-19 Care** | | | | | | | |
| No | 1,448 (21.0) | 636 (29.4) | 378 (20.4) | 199 (12.7) | 54 (5.5) | 181 (54.9) | <0.001 |
| All Services provided including COVID-19 care | 5,254 (76.3) | 1,456 (67.4) | 1,425 (77.0) | 1,325 (84.8) | 907 (92.6) | 141 (42.7) | |
| Only COVID-19 care provided | 182 (2.6) | 68 (3.1) | 48 (2.6) | 39 (2.5) | 19 (1.9) | 8 (2.4) | |
| **Treated COVID19 cases** | | | | | | | |
| No | 3,001 (43.6) | 1,557 (72.1) | 890 (48.1) | 347 (22.2) | 45 (4.6) | 162 (49.1) | <0.001 |
| Yes | 3,883 (56.4) | 603 (27.9) | 961 (51.9) | 1,216 (77.8) | 935 (95.4) | 168 (50.9) | |
| **Change in Patient Volume** | | | | | | | |
| No change | 1,636 (23.8) | 735 (34.0) | 466 (25.2) | 261 (16.7) | 101 (10.3) | 73 (22.1) | <0.001 |
| Patient volume INCREASED | 1,010 (14.7) | 325 (15.1) | 254 (13.7) | 239 (15.3) | 141 (14.4) | 51 (15.5) | |
| Patient Volume DECREASED | 4,238 (61.6) | 1,100 (50.9) | 1,131 (61.1) | 1,063 (68.0) | 738 (75.3) | 206 (62.4) | |
| **Source of Guidelines for COVID-19 from level of governance[b]** | | | | | | | |
| No Known Guidelines | 421 (6.1) | 204 (9.4) | 101 (5.5) | 52 (3.3) | 17 (1.7) | 47 (14.2) | <0.001 |
| Ministry of Health (National) | 5,017 (72.9) | 1,393 (64.5) | 1,318 (71.2) | 1,249 (79.9) | 900 (91.8) | 157 (47.6) | <0.001 |
| Regional | 1,351 (19.6) | 389 (18.0) | 384 (20.8) | 401 (25.7) | 137 (14.0) | 40 (12.12) | <0.001 |
| District | 1,823 (26.5) | 684 (31.7) | 598 (32.3) | 366 (23.4) | 85 (8.7) | 90 (27.3) | <0.001 |
| Municipality/Village | 582 (8.5) | 198 (9.2) | 153 (8.3) | 140 (9.0) | 34 (3.5) | 57 (17.3) | <0.001 |

a: Clinics are external to the general health system of the country and primary, secondary, and tertiary care.

b: HCW could report knowledge of guidelines originating from multiple sources.

**Table 2. Frequency and percent of reported COVID-19 IPC guideline implementation.**

| | Total N (%) | Dispensary N (%) | HC N (%) | Hospital N (%) | Referral Hospital N (%) | Clinics N (%) | P-Value* |
|---|---|---|---|---|---|---|---|
| **Personal Protection Policies** | | | | | | | |
| PPE availability during pandemic | 3,277 (47.6) | 900 (41.7) | 906 (48.9) | 834 (53.4) | 521 (53.2) | 116 (35.1) | <0.001 |
| Management of contaminated waste related to COVID-19 | 2,956 (42.9) | 800 (37.1) | 791 (42.7) | 770 (49.3) | 473 (48.3) | 122 (36.0) | <0.001 |
| Masking HCW and clients | 5,716 (83.1) | 1,709 (79.1) | 1,558 (84.2) | 1,335 (85.4) | 840 (85.7) | 274 (83.0) | <0.001 |
| **Care Changes in relation to COVID-19 Prevention regulations** | | | | | | | |
| Prioritization care for comorbidities | 3,827 (55.6) | 1,064 (49.3) | 1,065 (57.5) | 962 (61.5) | 587 (59.9) | 149 (45.2) | <0.001 |
| Implement appointment to reduce crowding | 2,802 (40.7) | 767 (35.5) | 774 (41.8) | 700 (44.8) | 46.3 (47.2) | 98 (29.7) | <0.001 |
| Virtual visits | 1,700 (24.7) | 489 (22.6) | 485 (26.2) | 411 (26.3) | 245 (25.0) | 70 (21.2) | 0.021 |
| Postpone routine medical and laboratory visits | 929 (13.5) | 208 (9.6) | 274 (14.8) | 251 (16.1) | 167 (17.1) | 29 (8.8) | <0.001 |
| Scaling up multi-month prescriptions | 2,951 (42.9) | 801 (37.1) | 870 (47.0) | 718 (45.9) | 483 (49.3) | 79 (23.9) | <0.001 |
| Min the number of people escorting patients | 3,625 (52.7) | 944 (43.7) | 1,013 (54.7) | 927 (59.3) | 603 (61.5) | 138 (41.8) | <0.001 |
| **COVID-19 Treatment protocols** | | | | | | | |
| Definition of suspected COVID-19 cases | 3,071 (44.6) | 864 (40.0) | 861 (46.5) | 781 (50.0) | 448 (45.7) | 117 (35.4) | <0.001 |
| COVID-19 triage | 2,973 (43.2) | 722 (33.4) | 819 (44.3) | 788 (50.4) | 549 (56.0) | 95 (28.8) | <0.001 |
| Collection of specimens for Laboratory Diagnosis | 2,437 (35.4) | 265 (16.9) | 584 (31.6) | 821 (52.5) | 585 (59.7) | 82 (24.8) | <0.001 |
| COVID-19 treatment protocol | 2,842 (41.3) | 561 (26.0) | 743 (40.1) | 863 (55.2) | 570 (58.2) | 105 (31.8) | <0.001 |
| Syndrome treatment and severe case management | 2,604 (37.8) | 510 (23.6) | 705 (38.1) | 767 (49.1) | 531 (54.2) | 91 (27.6) | <0.001 |
| Referral pathway for all COVID-19 cases | 2,513 (36.5) | 731 (33.8) | 773 (41.8) | 637 (40.7) | 269 (27.5) | 103 (31.2) | <0.001 |
| Referral pathway for all COVID-19 cases if severity require more care | 2,353 (34.2) | 526 (24.3) | 693 (37.4) | 691 (44.2) | 358 (36.5) | 85 (25.8) | <0.001 |
| Discharge criteria | 2,055 (29.9) | 308 (14.3) | 524 (28.3) | 641 (41.0) | 519 (53.0) | 63 (19.1) | <0.001 |

Note

* p-values are based on Pearson's Chi Square test.

hospitals (55.2%) and referral hospitals (58.2%) than in dispensaries (26.0%) or even clinics (31.8%) or health centers (40.1%). The age of health workers was strongly associated with an increase in reported COVID-19 policy implementation (Table 3). Specifically, those aged 50 years and over were more likely to report COVID-19 guidelines implementation compared to the reference group of 18–29 years (total score coefficient 2.36 CI: 1.68, 3.03) and those aged 30–49 years (total score coefficient 1.11, CI: 0.83, 1.38). Compared with the reference group of nurses, physicians had higher scores for the combined score (1.12 CI: 0.74, 1.51), while clinical officers had higher scores for COVID-19 triage and treatment protocols. Pharmacists had lower scores for all COVID-19 policies (-0.79 CI: -1.40, -0.19), general policies (-0.21 CI: -0.31, -0.10), and changes in care protocols to control and prevent COVID-19 (-0.27 CI: -0.47, -0.96). Compared with the reference group of dispensaries, total scores for guidelines implementation were slightly higher for respondents working in health centres (1.03 CI: 0.70, 1.35), hospitals (1.55 CI: 1.04, 2.06), referral hospitals (1.68 CI: 1.02, 2.34) and clinics (0.80 CI: 0.20, 1.43). Facilities designated to provide COVID-19 care services were associated with higher overall scores (0.91 CI: 0.76, 1.38) and higher scores for changes in care and COVID-19 triage and treatment. Compared to facilities with no known cases, facilities with confirmed COVID-19 cases had higher overall scores (0.45 CI: 0.068, 0.84). Knowledge of infection, prevention and control guidelines was associated with higher overall scores for guidelines issued by the MOH (1.91 CI: 1.62, 2.20), regional (1.98 CI: 1.64, 2.32), district (2.07 CI: 1.73, 2.41) and municipal authorities (0.78 CI: 0.43, 1.12), as well as higher scores for each of the three separate thematic categories.

**Table 3. Multivariate linear regression models of predictors of implementation of guidelines for the prevention and treatment of COVID-19 within the health facility.**

| | All COVID Policies Score 0–17 | | General Precaution Policies including PPE Score 0–3 | | Care Changes for COVID-19 Prevention and Control Score 0–6 | | COVID-19 triage and treatment protocols Score 0–8 | |
|---|---|---|---|---|---|---|---|---|
| Female | 0.042 | [-0.17,0.26] | -0.0079 | [-0.058,0.042] | 0.082 | [-0.0052,0.17] | -0.032 | [-0.14,0.078] |
| Age | | | | | | | | |
| 18–29 | Ref. | | Ref. | | Ref. | | Ref. | |
| 30–49 | 1.11*** | [0.83,1.38] | 0.16*** | [0.10,0.21] | 0.31*** | [0.20,0.41] | 0.64*** | [0.48,0.80] |
| 50+ | 2.36*** | [1.68,3.03] | 0.35*** | [0.24,0.46] | 0.60*** | [0.32,0.88] | 1.40*** | [1.04,1.77] |
| Health Worker Cadre | | | | | | | | |
| Nurse | Ref. | | Ref. | | Ref. | | Ref. | |
| Clinical Officer | 0.31 | [-0.028,0.64] | 0.011 | [-0.064,0.086] | 0.095 | [-0.031,0.22] | 0.20* | [0.032,0.37] |
| Medical Doctor | 1.12*** | [0.74,1.51] | 0.12* | [0.0054,0.24] | 0.29*** | [0.16,0.43] | 0.71*** | [0.50,0.92] |
| Technician | 0.58** | [0.17,1.00] | 0.097* | [0.0074,0.19] | 0.15 | [-0.0068,0.31] | 0.34** | [0.098,0.58] |
| Pharmacist | -0.79* | [-1.40,-0.19] | -0.21*** | [-0.31,-0.10] | -0.27* | [-0.47,-0.068] | -0.32 | [-0.65,0.015] |
| Other | -0.35* | [-0.69,-0.012] | 0.032 | [-0.049,0.11] | -0.052 | [-0.20,0.095] | -0.33** | [-0.53,-0.14] |
| Urban | 0.21 | [-0.21,0.64] | 0.0068 | [-0.070,0.084] | 0.064 | [-0.10,0.23] | 0.14 | [-0.066,0.35] |
| Health Facility Level | | | | | | | | |
| Dispensary | Ref. | | Ref. | | Ref. | | Ref. | |
| Health Centre | 1.03*** | [0.70,1.35] | 0.15*** | [0.071,0.23] | 0.26*** | [0.14,0.37] | 0.62*** | [0.45,0.79] |
| Hospital | 1.55*** | [1.04,2.06] | 0.25*** | [0.15,0.35] | 0.26** | [0.086,0.43] | 1.04*** | [0.76,1.33] |
| Referral Hospital | 1.68*** | [1.02,2.34] | 0.30*** | [0.16,0.44] | 0.33** | [0.13,0.52] | 1.05*** | [0.69,1.41] |
| Clinic | 0.80* | [0.20,1.40] | 0.17** | [0.045,0.31] | 0.17 | [-0.019,0.36] | 0.45** | [0.12,0.79] |
| Facility Ownership | | | | | | | | |
| Government | Ref. | | Ref. | | Ref. | | Ref. | |
| NGO or religious | -0.013 | [-0.47,0.44] | 0.0021 | [-0.11,0.11] | -0.089 | [-0.31,0.13] | 0.074 | [-0.16,0.30] |
| Private | -0.20 | [-0.56,0.17] | -0.070 | [-0.14,0.0058] | -0.070 | [-0.20,0.060] | -0.055 | [-0.25,0.14] |
| Designated for COVID Care Services | 0.91*** | [0.60,1.23] | 0.058 | [-0.0090,0.13] | 0.39*** | [0.28,0.51] | 0.46*** | [0.28,0.64] |
| COVID-19 Cases | | | | | | | | |
| No Cases | Ref. | | Ref. | | Ref. | | Ref. | |
| Suspected | 0.31 | [-0.046,0.66] | -0.099** | [-0.17,-0.029] | 0.089 | [-0.048,0.23] | 0.32** | [0.13,0.51] |
| Some or All Cases Confirmed | 0.45* | [0.068,0.84] | -0.080* | [-0.15,-0.0056] | 0.056 | [-0.067,0.18] | 0.48*** | [0.25,0.70] |
| Source of Guidelines for COVID-19 from level of governance | | | | | | | | |
| None | Ref. | | Ref. | | Ref. | | Ref. | |
| Ministry of Health (National) | 1.91*** | [1.62,2.20] | 0.28*** | [0.22,0.33] | 0.99*** | [0.86,1.12] | 0.64*** | [0.50,0.79] |
| Regional | 1.98*** | [1.64,2.32] | 0.37*** | [0.31,0.44] | 0.59*** | [0.42,0.76] | 1.01*** | [0.85,1.17] |
| District | 2.07*** | [1.73,2.41] | 0.36*** | [0.29,0.43] | 0.94*** | [0.82,1.05] | 0.78*** | [0.58,0.98] |
| Municipal/Village | 0.78*** | [0.43,1.12] | 0.15** | [0.055,0.25] | 0.23* | [0.044,0.42] | 0.40*** | [0.19,0.60] |

95% confidence intervals in brackets

* $p < 0.05$

** $p < 0.01$

*** $p < 0.001$.

## Challenges faced by HCWs

HCWs reported facing the following challenges:

- **Stress related to risks of infections:** 73.0% of HCWs experienced stress related to working with infectious diseases, primarily at hospitals (75.9%) and referral hospitals (74.0%) (Fig 1

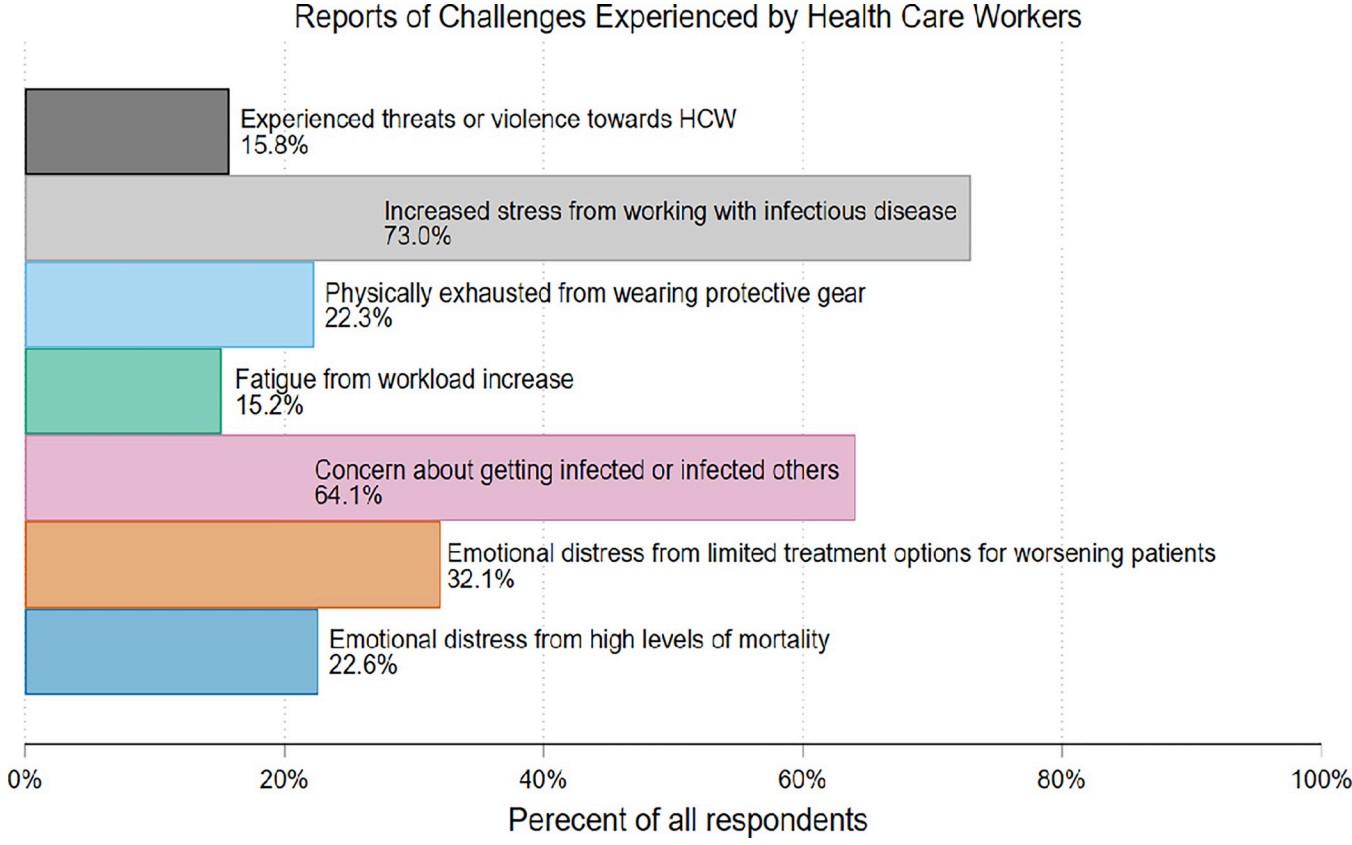

**Fig 1. Percentage of health workers reporting adversity since the beginning of COVID-19 –March 2020.**

and Table B in S1 Appendix). Among these, 64.1% expressed concern about their own infection or the risk of transmitting the disease with the majority at hospitals (69.1%) and referral hospitals (67.5%).

- **Emotional distress from limited treatment options:** 32.1% of respondents reported emotional distress due to limited treatment options for deteriorating patients, with the highest incidence recorded at referral hospitals (40.8%).

- **Emotional distress from high mortality:** Additionally, 22.6% reported emotional distress due to the increased mortality rate, primarily at hospitals and referral hospitals (26.6%).

- **Exhaustion from protective equipment:** Wearing additional protective equipment led to exhaustion in 22.3% of HCWs, most notably at hospitals (25.1%) and clinics (23.9%).

- **Fatigue from increased workload:** Furthermore, 15.2% of HCWs experienced fatigue due to an increased workload, primarily at referral hospitals (20.6%).

- **Threats and violence:** A concerning 15.8% of HCWs reported experiencing threats or violence, with the highest incidents reported at referral hospitals (21.2%) (Fig 1 and Table B in S1 Appendix).

   Variations among HCWs:

- **Gender Differences:** Female HCWs had lower odds of reporting stress from working with an infectious disease (0.74 CI: 0.68, 0.81) and exhaustion from wearing protective equipment

(0.79 CI: 0.69, 0.90), but higher odds of reporting worry about the uncertainty of becoming infected or infecting others (1.25 CI: 1.10, 1.42), as well as emotional distress from powerlessness when patients deteriorate (1.34 CI: 1.12, 1.49) or high mortality (1.31 CI: 1.17, 1.46) (Table 4).

- **Occupation differences:** Doctors reported higher odds of experiencing most of the challenges we specifically asked about, including threats or violence against HCWs (1.71 CI: 1.37, 2.13), exhaustion from wearing protective gear (2.02 CI: 1.58, 2.59), fatigue from increased workload (1.32 CI: 1.09, 1.59), fear of infection (1.24 CI: 1.06, 1.46), and emotional distress from high mortality (1.33 CI: 1.13, 1.56).

- **Urban vs. rural differences:** HCWs in urban facilities also had higher odds of many of the challenges listed, including violence against HCWs (1.27 CI: 1.06, 1.51), exhaustion from wearing protective gear (1.27 CI: 1.08, 1.49), fatigue from increased workload (1.33 CI: 1.10, 1.61), emotional distress from feeling powerless when patients deteriorate (1.15 CI: 1.01, 1.30), and emotional distress from high mortality (1.31 CI: 1.13, 1.50).

- **Health facility level:** There were minimal differences in the challenges faced by HCWs across health facility levels, except for the fear of becoming infected or infecting others, where the odds of reporting this challenge were higher in health centers (1.34 CI: 1.17, 1.53), hospitals (1.57 CI: 1.32, 1.87), and referral hospitals (1.50 CI: 1.10, 2.05) compared with dispensaries (Table 4).

### HCWs perceptions of the health system response to COVID-19

In terms of HCWs' views on the health system's response to COVID-19, there were high levels of agreement and strong agreement on questions related to the clarity and sources of information on COVID-19 guidelines, as well as the duty of HCWs to protect patients, even at the risk of their own and their families' health. In addition, 77% of HCWs believe that facility management cares about their well-being (Fig 2).

However, some questions received more mixed responses, with HCWs showing varying degrees of agreement and disagreement. Most HCWs felt that the policies in place to protect them were adequate (67%) and that they felt safe at work (58%). However, 53% reported a lack of personal protective equipment (PPE). Few HCWs reported fear of violence (23%), while 53% of them reported increased stress and concern about putting their families at risk because of their work (Fig 2).

The survey results suggest that HCWs received clear guidance (89%) and were strongly committed to the well-being of patients (69%). However, 47% of HCWs felt that health facilities were ill-prepared to handle COVID-19 and 43% reported concerns about the lack of personal protective equipment (PPE). Although many HCWs feel supported by their facility management and are committed to protecting patients, (53%) reported that their concerns about COVID-19 treatment have not been adequately addressed (Fig 2).

The outcome of a comprehensive multivariate regression analysis, examining the responses to Likert-type questions, indicates that physicians expressed a less favorable view regarding HCW protection, welfare, COVID-19 treatment, and policy in comparison to the reference group of nurses. This pattern holds true for both the overall assessment and thematic scales, as presented in Table 5.

Conversely, technicians, pharmacists and other HCW cadres had higher scores on all scales. This finding suggests that the different level of responsibility and challenges faced by doctors compared to other HCWs may influence their opinions. In addition, clinical officers had

**Table 4. Multivariate logistic regression models of predictors of reported challenges faced by HCW.**

| | Has experienced violence or threats to HCW | stress from working with an infectious disease | exhausted from wearing protective gear | fatigue from workload increases | concerns of getting infected or infecting others | emotional distress from limited treatment options | emotional distress from high levels of mortality |
|---|---|---|---|---|---|---|---|
| Female | 0.99 | 0.74*** | 0.79*** | 0.98 | 1.25*** | 1.34*** | 1.31*** |
| | [0.86,1.14] | [0.68,0.81] | [0.69,0.90] | [0.83,1.15] | [1.10,1.42] | [1.21,1.49] | [1.17,1.46] |
| 18–29 | Ref | Ref | Ref | Ref | Ref | Ref | Ref |
| 30–49 | 1.11 | 1.22* | 1.12 | 0.91 | 1.22** | 1.15 | 1.14* |
| | [0.95,1.30] | [1.02,1.46] | [0.92,1.35] | [0.78,1.05] | [1.06,1.39] | [1.00,1.34] | [1.00,1.29] |
| 50+ | 1.09 | 1.29 | 1.55* | 1.46* | 1.57* | 1.01 | 1.21 |
| | [0.79,1.52] | [0.96,1.74] | [1.10,2.20] | [1.04,2.04] | [1.01,2.44] | [0.76,1.34] | [0.85,1.72] |
| Health Worker Cadre | | | | | | | |
| Nurse | Ref | Ref | Ref | Ref | Ref | Ref | Ref |
| Clinical Officer | 1.20 | 1.03 | 0.99 | 0.91 | 1.17* | 0.93 | 0.97 |
| | [0.98,1.45] | [0.91,1.17] | [0.83,1.19] | [0.75,1.10] | [1.01,1.35] | [0.78,1.09] | [0.81,1.16] |
| Medical Doctor | 1.71*** | 1.03 | 2.02*** | 1.32** | 1.24** | 1.15 | 1.33*** |
| | [1.37,2.13] | [0.88,1.22] | [1.58,2.59] | [1.09,1.59] | [1.06,1.46] | [0.97,1.38] | [1.13,1.56] |
| Technician | 0.93 | 0.98 | 1.08 | 1.03 | 1.03 | 1.23 | 1.14 |
| | [0.72,1.19] | [0.83,1.15] | [0.89,1.31] | [0.80,1.31] | [0.89,1.19] | [0.97,1.56] | [0.94,1.39] |
| Pharmacist | 1.03 | 0.93 | 1.59*** | 1.29 | 0.87 | 0.90 | 0.94 |
| | [0.76,1.38] | [0.69,1.26] | [1.22,2.08] | [0.92,1.82] | [0.66,1.15] | [0.72,1.13] | [0.72,1.24] |
| Other | 0.75* | 1.16 | 2.03*** | 0.77 | 0.80* | 0.60*** | 0.76** |
| | [0.58,0.99] | [0.95,1.42] | [1.64,2.52] | [0.59,1.00] | [0.66,0.97] | [0.48,0.74] | [0.62,0.93] |
| Urban | 1.27** | 0.99 | 1.27** | 1.33** | 1.13 | 1.15* | 1.31*** |
| | [1.06,1.51] | [0.87,1.14] | [1.08,1.49] | [1.10,1.61] | [0.97,1.31] | [1.01,1.30] | [1.13,1.50] |
| Health Facility Level | | | | | | | |
| Dispensary | Ref | Ref | Ref | Ref | Ref | Ref | Ref |
| Health Centre | 1.12 | 1.11 | 0.86** | 0.98 | 1.34*** | 1.05 | 0.97 |
| | [0.96,1.31] | [0.94,1.31] | [0.77,0.95] | [0.78,1.23] | [1.17,1.53] | [0.88,1.27] | [0.81,1.16] |
| Hospital | 0.85 | 1.15 | 0.94 | 1.14 | 1.57*** | 1.34** | 1.04 |
| | [0.67,1.07] | [0.93,1.41] | [0.79,1.12] | [0.85,1.53] | [1.32,1.87] | [1.10,1.63] | [0.86,1.25] |
| Referral Hospital | 0.88 | 1.07 | 0.81 | 1.10 | 1.50* | 1.27* | 0.97 |
| | [0.70,1.10] | [0.87,1.31] | [0.62,1.05] | [0.76,1.59] | [1.10,2.05] | [1.03,1.57] | [0.80,1.18] |
| Clinic | 1.69* | 0.98 | 0.88 | 1.09 | 1.10 | 0.95 | 1.11 |
| | [1.10,2.60] | [0.76,1.26] | [0.67,1.17] | [0.67,1.76] | [0.82,1.48] | [0.71,1.26] | [0.82,1.50] |
| Facility Ownership | | | | | | | |
| Government | Ref | Ref | Ref | Ref | Ref | Ref | Ref |
| NGO/Religious | 0.73** | 1.01 | 1.48*** | 1.58*** | 1.07 | 1.28* | 1.38** |
| | [0.59,0.89] | [0.84,1.22] | [1.23,1.80] | [1.25,2.00] | [0.90,1.27] | [1.05,1.57] | [1.11,1.71] |
| Private | 0.71** | 0.99 | 1.04 | 0.90 | 0.97 | 1.35** | 1.13 |
| | [0.56,0.91] | [0.84,1.16] | [0.80,1.37] | [0.69,1.17] | [0.82,1.16] | [1.13,1.61] | [0.96,1.33] |
| Designated for COVID-19 Care | 1.26* | 1.09 | 0.83* | 0.64*** | 1.10 | 1.23* | 1.17* |
| | [1.01,1.55] | [0.96,1.24] | [0.72,0.97] | [0.54,0.75] | [0.97,1.25] | [1.05,1.44] | [1.01,1.36] |
| COVID-19 Cases | | | | | | | |
| No Cases | Ref | Ref | Ref | Ref | Ref | Ref | Ref |
| Suspected | 1.86*** | 1.12 | 1.14 | 1.62*** | 0.98 | 1.35*** | 1.32*** |
| | [1.54,2.24] | [0.94,1.33] | [0.93,1.40] | [1.33,1.98] | [0.87,1.11] | [1.14,1.59] | [1.12,1.56] |

(*Continued*)

**Table 4.** (Continued)

| | Has experienced violence or threats to HCW | stress from working with an infectious disease | exhausted from wearing protective gear | fatigue from workload increases | concerns of getting infected or infecting others | emotional distress from limited treatment options | emotional distress from high levels of mortality |
|---|---|---|---|---|---|---|---|
| Some or All Cases Confirmed | 2.01*** | 1.20* | 1.07 | 2.19*** | 1.02 | 1.72*** | 1.52*** |
| | [1.61,2.50] | [1.01,1.41] | [0.86,1.33] | [1.69,2.84] | [0.89,1.18] | [1.49,1.99] | [1.30,1.77] |
| Source of Guidelines for COVID-19 from level of governance | | | | | | | |
| No Guidelines | Ref | Ref | Ref | Ref | Ref | Ref | Ref |
| Ministry of Health (National) | 1.07 | 1.20** | 1.10 | 1.14 | 1.31*** | 1.19** | 1.18 |
| | [0.89,1.29] | [1.05,1.37] | [0.94,1.28] | [0.97,1.34] | [1.16,1.48] | [1.05,1.34] | [0.99,1.40] |
| Regional | 1.20* | 1.50*** | 1.43*** | 1.52*** | 1.36*** | 1.29*** | 1.30** |
| | [1.04,1.37] | [1.25,1.80] | [1.23,1.66] | [1.24,1.86] | [1.16,1.60] | [1.12,1.48] | [1.08,1.56] |
| District | 0.90 | 1.04 | 1.02 | 1.12 | 1.60*** | 1.13 | 1.15 |
| | [0.73,1.10] | [0.91,1.20] | [0.87,1.21] | [0.92,1.37] | [1.35,1.89] | [0.96,1.32] | [0.97,1.37] |
| Municipal/ Village | 0.83 | 1.15 | 1.12 | 1.49** | 1.39** | 1.29* | 1.44*** |
| | [0.65,1.06] | [0.92,1.44] | [0.83,1.51] | [1.12,1.97] | [1.10,1.75] | [1.02,1.62] | [1.18,1.75] |

Reported coefficients represent odds rations with 95% confidence intervals in brackets.

* $p < 0.05$

** $p < 0.01$

*** $p < 0.001$.

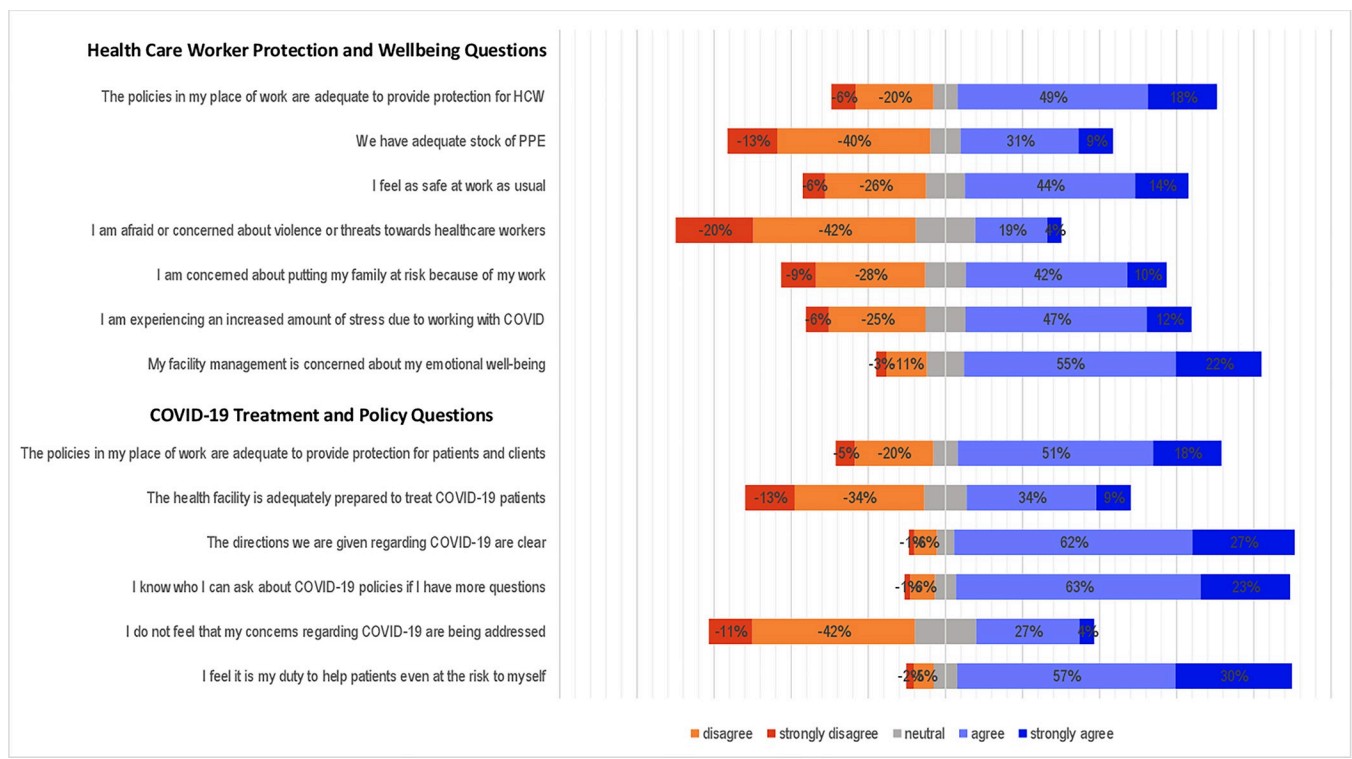

**Fig 2. Health Care Worker Perceptions on health system response to Covid-19.**

**Table 5. Multivariate logistic regression models of likert scales of HCW perception of protection and well-being.**

| | Scale from all Likert style questions | | Scale from HCW protection and wellbeing questions | | Scale from COVID-19 treatment and policy questions | |
|---|---|---|---|---|---|---|
| Female | 0.021 | [-0.61,0.65] | -0.093 | [-0.47,0.28] | 0.11 | [-0.16,0.39] |
| Age | | | | | | |
| 18–29 | Ref. | | Ref. | | Ref. | |
| 30–49 | -1.61*** | [-2.16,-1.07] | -0.98*** | [-1.30,-0.66] | -0.63*** | [-0.87,-0.40] |
| 50+ | -0.25 | [-1.37,0.86] | -0.23 | [-0.93,0.47] | -0.019 | [-0.52,0.48] |
| Health Worker Cadre | | | | | | |
| Nurse | Ref. | | Ref. | | Ref. | |
| Clinical Officer | -0.35 | [-0.84,0.13] | -0.36* | [-0.66,-0.067] | 0.0068 | [-0.24,0.25] |
| Medical Doctor | -2.06*** | [-2.54,-1.57] | -1.24*** | [-1.56,-0.92] | -0.82*** | [-1.05,-0.60] |
| Technician | 1.54*** | [0.79,2.29] | 1.08*** | [0.67,1.49] | 0.46* | [0.065,0.86] |
| Pharmacist | 1.78*** | [0.87,2.69] | 1.18*** | [0.66,1.70] | 0.60* | [0.14,1.06] |
| Other | 1.33** | [0.57,2.09] | 0.87*** | [0.40,1.35] | 0.46** | [0.12,0.79] |
| Urban | 0.24 | [-0.59,1.06] | -0.0029 | [-0.50,0.49] | 0.24 | [-0.11,0.58] |
| Health Facility Level | | | | | | |
| Dispensary | Ref. | | Ref. | | Ref. | |
| Health Centre | -0.78** | [-1.25,-0.30] | -0.50** | [-0.81,-0.20] | -0.27* | [-0.50,-0.043] |
| Hospital | -0.77 | [-1.69,0.15] | -0.59* | [-1.17,-0.022] | -0.18 | [-0.57,0.21] |
| Referral Hospital | -0.57 | [-2.03,0.89] | -0.53 | [-1.41,0.34] | -0.036 | [-0.65,0.58] |
| Private Clinic | 0.42 | [-0.53,1.38] | 0.050 | [-0.56,0.66] | 0.37 | [-0.056,0.81] |
| Facility Ownership | | | | | | |
| Government | Ref. | | Ref. | | Ref. | |
| NGO or religious | 1.50** | [0.57,2.42] | 0.92** | [0.38,1.47] | 0.57* | [0.099,1.04] |
| Private | 1.81*** | [1.19,2.43] | 1.08*** | [0.68,1.49] | 0.73*** | [0.45,1.00] |
| Designated for COVID-19 Care Services | 1.80*** | [1.29,2.31] | 0.70*** | [0.38,1.03] | 1.10*** | [0.87,1.32] |
| COVID-19 Cases | | | | | | |
| No Cases | Ref. | | Ref. | | Ref. | |
| Suspected | -0.47* | [-0.91,-0.026] | -0.52*** | [-0.81,-0.24] | 0.053 | [-0.17,0.27] |
| Some or All Cases Confirmed | 0.96*** | [0.46,1.47] | 0.11 | [-0.21,0.43] | 0.85*** | [0.61,1.09] |
| Source of Guidelines for COVID-19 from level of governance | | | | | | |
| None | Ref. | | Ref. | | Ref. | |
| Ministry of Health (National) | 0.95** | [0.39,1.51] | 0.28 | [-0.036,0.60] | 0.67*** | [0.40,0.93] |
| Regional | -0.33 | [-1.05,0.40] | -0.30 | [-0.70,0.10] | -0.028 | [-0.41,0.36] |
| District | -0.38 | [-0.93,0.17] | -0.39* | [-0.78,-0.0094] | 0.011 | [-0.22,0.24] |
| Municipal/Village | 1.10* | [0.27,1.93] | 0.52* | [0.076,0.96] | 0.58* | [0.12,1.05] |
| Constant | 42.7*** | [41.7,43.7] | 22.5*** | [21.9,23.1] | 20.2*** | [19.7,20.7] |
| Observations | 6878 | | 6878 | | 6878 | |

95% confidence intervals in brackets

* $p < 0.05$

** $p < 0.01$

*** $p < 0.001$.

lower scores on the HCW protection and well-being scale. Facility ownership also had an impact, with NGO/religious organisation and privately-owned facilities scoring higher than government-owned facilities on all three scales. Facilities that were designated to provide COVID-19 services had higher scores on all scales than those that were not. However, facilities with suspected COVID-19 cases had lower scores on the total scale and the HCW protection

and well-being scale, whereas facilities with confirmed cases had higher scores on both the total scale and the COVID-19 treatment and policy questions scale. In addition, awareness of guidelines issued by the MOH was associated with higher scores on the total scale and the COVID-19 treatment and policy questions scale. Facilities that were aware of guidelines issued at the community or village level also had higher scores on all three scales. Finally, logistic regression for each Likert-type question, when converted into binary variables for "agree/strongly agree" versus other responses, further illustrates the trends for each question in the scales (Table C in S1 Appendix and Table D in S1 Appendix).

## Discussion

This study is, to the best of our knowledge, the first to examine the health system's response to the pandemic from the perspective of HCWs in Tanzania. We collected primary data through an online questionnaire distributed to HCWs via WhatsApp groups. This is a novel and promising approach in health policy and systems research [37]. The questionnaire included questions related to the implementation of the COVID-19 protocols, HCWs' perceptions of protection and well-being, and concerns related to the treatment of COVID-19.

These findings suggest that different factors, such as HCW cadre, facility ownership and COVID-19 designation status, influence HCWs' opinions about the health system's response to COVID-19. These findings could inform targeted interventions to improve HCW protection and well-being, and management and policy in health facilities for COVID-19 or future health emergencies.

The results of the study show that the majority of HCWs were aware of all the guidelines issued to control COVID-19, in particular those issued by the MOH. However, consistent with previous studies, the study highlights the variability in the implementation of COVID-19 protocols at different levels of health facilities [4,19,38,39]. HCWs in Tanzania reported heightened stress levels attributed to their work with infectious diseases. This stress stems from concerns about contracting infections themselves or transmitting them to others, largely due to the inadequacies in pandemic preparedness within the healthcare system. These findings align with prior studies that have documented deficiencies in health system preparedness, interruptions in healthcare services, and the psychological distress experienced by HCWs [40–42]. Mitigating these fears and addressing other related concerns should be regarded as a pivotal component of an effective pandemic response [31,43].

HCWs in urban areas were more likely to face challenges than those in rural areas, reflecting the faster spread of COVID-19 in urban areas and the higher demand for health care and protection in urban facilities [44,45]. This highlights the need for strong preventive measures in urban settings, including the provision of infrastructure to enable sustainable pandemic prevention [44,46]. Medical doctors were more likely to report a higher number of challenges, possibly reflecting the additional responsibility they have to manage and protect patients and staff. In addition, perceptions of protection and well-being varied widely among different HCW cadres. It is therefore important to implement targeted interventions based on the level of risk exposure of HCWs, rather than one-size-fits-all interventions [47]. These findings are consistent with finding from studies conducted in different settings in Africa, India and Iran, which have reported that different cadres of HCWs experience different levels of post-traumatic stress symptoms depending on their level of exposure [14,15,17,18,48]. However, the study did not find significant differences in perceptions of protection and well-being between different levels of health facilities. Notably, HCWs employed in government health facilities scored lower on both HCW protection and well-being and on questions related to COVID-19 treatment and policy. This may indicate a weaker capacity of government health facilities to

implement HCW protection measures compared to private facilities, or it may be related to different levels of exposure to COVID-19 patients that could not be captured in this study. The results of this study highlight the importance of supporting HCWs through the dissemination and implementation of guidelines, as well as social and emotional support to help them cope with the challenges they face. Interventions and policies should be developed to address the challenges identified and strengthen the capacity of the health system to respond to future pandemics. Ongoing training and capacity-building programmes for HCWs are also needed to ensure that they have the necessary knowledge and skills to respond effectively to pandemics. Such efforts could improve HCWs' perceptions of protection and well-being, thereby enhancing the resilience of the health system. By addressing these challenges, the health system in Tanzania can become more resilient and respond more effectively to future pandemics.

This study has some limitations. First, it is uncertain whether the sample analysed is truly representative of the wider population of HCWs in Tanzania. While we randomly selected from all Tanzanian districts, participation in the survey was voluntary. Overall, the study was able to achieve a substantial sample size in a short period of time through the use of mobile technology, which allowed HCWs to respond at their convenience. This also resulted in a wide sampling across HCW cadres and health facilities. Second, the study relies on self-reported measures, which may affect the accuracy of responses. This is particularly relevant in the context of experience with the COVID-19 guidelines, which may lead to biased results. Thirdly, the lack of accurate and geo-referenced data on COVID-19 cases in Tanzania limited our ability to validate some of the findings related to the level of exposure of HCWs to COVID-19 patients. Despite these limitations, the study provides valuable insights into the challenges faced by HCWs in Tanzania from their own perspective regarding the COVID-19 pandemic.

In conclusion, this study shows that HCWs in Tanzania have a high level of awareness and adherence to guidelines aimed at mitigating the impact of COVID-19 in their facilities. However, the study also reveals several challenges faced by HCWs, including increased stress, concerns about infection, inadequate personal protective equipment, and differences in perceptions of protection and well-being among different HCW cadres. These findings highlight the importance of consistent implementation of guidelines and social and emotional support for HCWs, as well as the need for targeted interventions based on exposure levels.

## Supporting information

**S1 Checklist. PLOS' questionnaire on inclusivity in global research.**
(DOCX)

**S1 Appendix.** Table A: Odds ratios for Likert style questions to be Agree/Strongly Agree Table B: Odds ratios for more Likert style questions to be Agree/Strongly Agree Table C: Reported Challenges from Healthcare workers Table D: Health care worker questionnaire. (DOCX)

## Acknowledgments

We would like to acknowledge the assistance of the President office, ministry of regional and local government administration (PO-RALG) in Dodoma Tanzania, regional and district managers for their support during the implementation of the study.

## Author Contributions

**Conceptualization:** Kassimu Tani, Brianna Osetinsky, Fabrizio Tediosi.

**Data curation:** Kassimu Tani, Grace Mhalu, Sally Mtenga.

**Formal analysis:** Kassimu Tani, Brianna Osetinsky, Günther Fink, Fabrizio Tediosi.

**Funding acquisition:** Fabrizio Tediosi.

**Methodology:** Kassimu Tani, Brianna Osetinsky, Fabrizio Tediosi.

**Writing – original draft:** Kassimu Tani, Brianna Osetinsky, Günther Fink, Fabrizio Tediosi.

**Writing – review & editing:** Kassimu Tani, Brianna Osetinsky, Grace Mhalu, Sally Mtenga, Günther Fink, Fabrizio Tediosi.

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
