## [Decision Letter · Decision Letter 0]

4 Sep 2023

PGPH-D-23-00842

Compliance and burnout: Healthcare workers' experiences with COVID-19-related prevention and control measures in Tanzania

Dear Dr. Tani,

Thank you for submitting your manuscript to PLOS Global Public Health. After careful consideration, we feel that it has merit but does not fully meet PLOS Global Public Health’s publication criteria as it currently stands. Therefore, we invite you to submit a revised version of the manuscript that addresses the points raised during the review process.

Your manuscript has been evaluated by two reviewers, and their comments are appended below.

The reviewers have commented on the data collection reported in this study, particularly regarding the validation and/or piloting process used in the development of the questionnaire used. Other comments relate to the data analyses chosen, as well as ensuring that the conclusions reported are directly related to the study data. Please ensure you address each of the reviewers' comments when revising your manuscript.

We look forward to receiving your revised manuscript.

Kind regards,

Hugh Cowley

Staff Editor

Journal Requirements:

1) Please include a complete copy of PLOS’ questionnaire on inclusivity in global research in your revised manuscript. Our policy for research in this area aims to improve transparency in the reporting of research performed outside of researchers’ own country or community. The policy applies to researchers who have travelled to a different country to conduct research, research with Indigenous populations or their lands, and research on cultural artefacts. The questionnaire can also be requested at the journal’s discretion for any other submissions, even if these conditions are not met.  Please find more information on the policy and a link to download a blank copy of the questionnaire here: https://journals.plos.org/globalpublichealth/s/best-practices-in-research-reporting. Please upload a completed version of your questionnaire as Supporting Information when you resubmit your manuscript.

2) Please provide additional details regarding participant consent. In the ethics statement in the Methods and online submission information, please ensure that you have specified (1) whether consent was informed and (2) what type you obtained (for instance, written or verbal, and if verbal, how it was documented and witnessed). If your study included minors, state whether you obtained consent from parents or guardians. If the need for consent was waived by the ethics committee, please include this information.

If you are reporting a retrospective study of medical records or archived samples, please ensure that you have discussed whether all data were fully anonymized before you accessed them and/or whether the IRB or ethics committee waived the requirement for informed consent. If patients provided informed written consent to have data from their medical records used in research, please include this information."

3. Please provide separate figure files in .tif or .eps format.

Additional Editor Comments (if provided):

Reviewers' comments:

Reviewer's Responses to Questions

**Comments to the Author**

1. Does this manuscript meet PLOS Global Public Health’s publication criteria? Is the manuscript technically sound, and do the data support the conclusions? The manuscript must describe methodologically and ethically rigorous research with conclusions that are appropriately drawn based on the data presented.

Reviewer #1: Yes

Reviewer #2: Partly

2. Has the statistical analysis been performed appropriately and rigorously?

Reviewer #1: Yes

Reviewer #2: No

3. Have the authors made all data underlying the findings in their manuscript fully available (please refer to the Data Availability Statement at the start of the manuscript PDF file)?

Reviewer #1: Yes

Reviewer #2: Yes

4. Is the manuscript presented in an intelligible fashion and written in standard English?

Reviewer #1: Yes

Reviewer #2: Yes

5. Review Comments to the Author

Reviewer #1: This manuscript presents an important and timely research topic that is the first of kind in not only Tanzania but in many sub-regions on the African continent. Healthcare workers (HCWs) experiences of implementing national and regional COVID-19 guidelines was certainly daunting in many parts of the world as research has shown but to capture and reveal what healthcare workers in sub-Saharan African HCWs experienced is novel by the authors and showcases the magnitude of what other HCWs encountered throughout the COVID-19 pandemic.

I have read the manuscript several times and have concluded that the authors conducted the study in a novel and appropriate manner adhering to research ethics guidelines and step-wise procedure. The methodology including the study design, settings, sampling, and data collection were thoroughly described in clear language. The probability sampling used to select the districts was appropriate even though the individual HCW sampling was not very clear. My concern in the data collection is whether the Enkato and ODK were adopted without modification or they were modified. Again, did the researchers pilot test the instruments prior to collecting data?

The statistical analysis in my opinion was robust and clearly presents the findings of the study that I believe many readers will understand and appreciate but I also believe that the write up of the results could be presented in a more step-by-step manners for most readers. The tables are presented very clearly nonetheless.

The discussion is well presented and reflects what the relationships of the findings are to existing literature. In addition, I would recommend adding examples of what other studies specifically reported rather than merely indicating that the findings are consistent or similar to what others have found.

Overall, this is great and timely study that sets the pace for other researchers in others sub-Saharan countries to replicate so that governments and healthcare authorities get to understand the treatment and guideline dynamics and the safety and well-being of HCWs in times of pandemics. This will help plan better and provide appropriate interventions for all cadres of HCWs.

Reviewer #2: Thank you for the opportunity to review this manuscript. The study aims to investigate the HCW:s experience of COVID-19 related prevention and control measures implemented in Tanzania.

The subject is interesting and there is a great need for systematically investigate and learn from the COVID-19 pandemic, especially in low-income countries. However, I have some major concerns, mainly regarding methods and the interpretation of the findings in this study, that needs some further efforts to entangle.

Major concerns

1. There is a discrepancy between the title, aim and conclusions in the manuscript which needs to be addressed. The title includes the terms “compliance” and “burnout” which has not been assessed in the study and should be removed from the title.

2. In this study, a project specific COVID-19 questionnaire was used (as in most similar studies). How was this questionnaire developed, tested a validated? The ability of the questionnaire to accurately measure the correct aspects within the study will highly affect the possibility interpret the findings correctly. The authors need to add information on how the questionnaire was developed, tested and validated.

3. In the data analysis, descriptive data has been calculated, differences have been investigated using chi-square tests and multivariate regression analysis have been conducted. The descriptive analyses have been described but the multivariate analyses need to be described more in detail to facilitate the reading and interpretation of the findings.

Firstly, the rationale for performing these analyses (“to examine the association between HCW and health facility characteristics and the composite implementation scores, challenges faced, and perception Likert scale scores”) is quite vague and could be explained more in detail.

Secondly, a long list of independent variables is presented, where all these included in the same model or where they used individually? Were the analyses adjusted for any of the background variables and why?

Thirdly, overall composite scores and separate composite scores were calculated and used in the analyses. The authors need to justify this and provide detailed information on whether these scores accurately reflect general precaution, changes in protocols to control etc. and how these scores may be interpreted (or refrain from using composite scores)

4. The interpretation of the findings needs to be revised in order to make sure that conclusions are drawn from the data at hand.

Firstly, the HCW:s situation during normal operations before the COVID-19 pandemic need to be acknowledge and how this will affect the findings in this study. I.e. the authors need to separate the effect from the pandemic from normal operations. For instance (but not limited to), on line 296-297, the authors state that “majority of HCWs reported fear of violence, while fewer HCWs reported increased stress and concern about putting their families at risk because of their work”. Looking at the questionnaire, 2 out of three questions supporting these findings are not specifically connected to the pandemic. How common is it for HCW:s to experiencing violence under normal operations? How should these results be interpreted in light of the aim of this study?

Secondly, on line 213-216 the authors state that there was a considerable variation in guideline implementation between different facilities. Is this difference caused by differences in implementation (as suggested by the author) or could it be caused by differences in the healthcare facilities (see table 1)? Is it also plausible to assume that all facilities would have 100% implementation on all guidelines regardless of the nature of their operations? All facilities might have the possibilities to use masks, but could all facilities postpone all routine visits, or will certain healthcare facilities have patient groups with for instance chronic diseases which requires medical surveillance regardless of the pandemic? How should these differences be interpreted and how will it affect the authors findings/conclusions?

Minor concerns

1. The background contains adequate information about the management of the pandemic in Tanzania but lacks information on the need of covid-care or the covid-19 disease burden during the time of the study which needs to be added in order to be able to compare the findings with other studies now and in the future.

2. Make sure that Table 1 contains all information necessary to be interpreted such as explaining that n (%) are presented in the table. Also, revise the table to make sure that the number of decimals is consistent throughout the table and whether to write 1000 or 1,000 for values above 999. P-values <0.000 should also be changes to <0.001

3. The abbreviation MOH (line 146) has not been explained in the text

4. On line 220 and onwards, results from the regression analyses are presented. It would benefit the reader to be clear about which factors that are analysed instead of writing “have a larger effect size compared to the reference group” etc.

5. Line 319-322 seems to contain a discussion of the result rather that presenting findings from the study. The authors should consider moving this part to the discussion section.

6. PLOS authors have the option to publish the peer review history of their article (what does this mean?). If published, this will include your full peer review and any attached files.

**Do you want your identity to be public for this peer review?** For information about this choice, including consent withdrawal, please see our Privacy Policy.

Reviewer #1: No

Reviewer #2: No

---

## [Decision Letter · Decision Letter 1]

23 Oct 2023

PGPH-D-23-00842R1

Healthcare workers' experiences with COVID-19-related prevention and control measures in Tanzania

Dear Dr. Tani,

Thank you for submitting your manuscript to PLOS Global Public Health. After careful consideration, we feel that it has merit but does not fully meet PLOS Global Public Health’s publication criteria as it currently stands. Therefore, we invite you to submit a revised version of the manuscript that addresses the points raised during the review process.

Please see a few very minor suggestions from one of the reviewers below.

We look forward to receiving your revised manuscript.

Kind regards,

Hanna Landenmark

Staff Editor

Journal Requirements:

Additional Editor Comments (if provided):

Reviewers' comments:

Reviewer's Responses to Questions

**Comments to the Author**

1. If the authors have adequately addressed your comments raised in a previous round of review and you feel that this manuscript is now acceptable for publication, you may indicate that here to bypass the “Comments to the Author” section, enter your conflict of interest statement in the “Confidential to Editor” section, and submit your "Accept" recommendation.

Reviewer #1: All comments have been addressed

Reviewer #2: All comments have been addressed

2. Does this manuscript meet PLOS Global Public Health’s publication criteria? Is the manuscript technically sound, and do the data support the conclusions? The manuscript must describe methodologically and ethically rigorous research with conclusions that are appropriately drawn based on the data presented.

Reviewer #1: Yes

Reviewer #2: Yes

3. Has the statistical analysis been performed appropriately and rigorously?

Reviewer #1: (No Response)

Reviewer #2: Yes

4. Have the authors made all data underlying the findings in their manuscript fully available (please refer to the Data Availability Statement at the start of the manuscript PDF file)?

Reviewer #1: No

Reviewer #2: Yes

5. Is the manuscript presented in an intelligible fashion and written in standard English?

Reviewer #1: Yes

Reviewer #2: Yes

6. Review Comments to the Author

Reviewer #1: All comments that I made in my last review have been adequately address. The only new thing I have observed that the authors should try and present the result on challenges in categories to make them more clearer to the reader. It appears that the reported challenges faced by the HCWs are lumped together.

Lines 309-313 talk about HCWs perception of the health system response where the authors used terms such as agreements and strong agreement without specifying exactly what HCWs agreed on. Also, the use of MOST without indicating the necessary proportions or percentages. Please, clarify these.

In all, I think the authors addressed the comments and strengthened the manuscripts quality and originality.

Reviewer #2: (No Response)

7. PLOS authors have the option to publish the peer review history of their article (what does this mean?). If published, this will include your full peer review and any attached files.

**Do you want your identity to be public for this peer review?** For information about this choice, including consent withdrawal, please see our Privacy Policy.

Reviewer #1: No

Reviewer #2: No

---

## [Editor Report · Decision Letter 2]

8 Nov 2023

Healthcare workers' experiences with COVID-19-related prevention and control measures in Tanzania

PGPH-D-23-00842R2

Dear Mr Tani,

We are pleased to inform you that your manuscript 'Healthcare workers' experiences with COVID-19-related prevention and control measures in Tanzania' has been provisionally accepted for publication in PLOS Global Public Health.

Best regards,

Julia Robinson

Executive Editor